# Multi-Omics Approach Reveals miR-SNPs Affecting Muscle Fatty Acids Profile in Nelore Cattle

**DOI:** 10.3390/genes12010067

**Published:** 2021-01-06

**Authors:** Tainã Figueiredo Cardoso, Luiz Lehmann Coutinho, Jennifer Jessica Bruscadin, Wellison Jarles da Silva Diniz, Juliana Petrini, Bruno Gabriel Nascimento Andrade, Priscila Silva Neubern de Oliveira, Mirele Daiana Poleti, Aline Silva Mello Cesar, Juliano Coelho da Silveira, Marcos Roberto Chiaratti, Adhemar Zerlotini, Gerson Barreto Mourão, Luciana Correia de Almeida Regitano

**Affiliations:** 1Embrapa Pecuária Sudeste, Rodovia Washington Luiz, Km 234, s/n, Fazenda Canchim, São Carlos, SP 339, Brazil; tainafcardoso@gmail.com (T.F.C.); bgabriel.andrade@gmail.com (B.G.N.A.); 2Department of Animal Science, “Luiz de Queiroz” College of Agriculture, University of São Paulo/ESALQ, Piracicaba, SP 13418-900, Brazil; llcoutinho@usp.br (L.L.C.); juliana.petrini@gmail.com (J.P.); gbmourao@usp.br (G.B.M.); 3Departamento de Genética e Evolução, Universidade Federal de São Carlos (UFSCar), São Carlos, SP 13565-905, Brazil; jennifer.j.bruscadin@gmail.com (J.J.B.); priscilaneuberndeoliveira@gmail.com (P.S.N.d.O.); marcos.chiaratti@ufscar.br (M.R.C.); 4Department of Animal Sciences, North Dakota State University, Fargo, ND 58105, USA; wjarles09@gmail.com; 5Department of Veterinary Medicine, Faculty of Animal Science and Food Engineering, University of São Paulo (USP), Pirassununga, SP 13635-900, Brazil; mdpoleti@gmail.com (M.D.P.); julianodasilveira@usp.br (J.C.d.S.); 6Department of Agri-Food Industry, Food and Nutrition, “Luiz de Queiroz” College of Agriculture, University of São Paulo/ESALQ, Piracicaba, SP 13418-900, Brazil; alinecesar@usp.br; 7Embrapa Informática Agropecuária, Campinas, SP 13083-886, Brazil; adhemar.zerlotini@embrapa.br

**Keywords:** polymorphism, association analysis, miRNAs, *Bos indicus*, beef quality

## Abstract

MicroRNAs (miRNAs) are key regulators of gene expression, potentially affecting several biological processes, whose function can be altered by sequence variation. Hence, the integration of single nucleotide polymorphisms (SNP) and miRNAs can explain individual differences in economic traits. To provide new insights into the effects of SNPs on miRNAs and their related target genes, we carried out a multi-omic analysis to identify SNPs in miRNA mature sequences (miR-SNPs) associated with fatty acid (FA) composition in the Nelore cattle. As a result, we identified 3 miR-SNPs in different miRNAs (bta-miR-2419-3p, bta-miR-193a-2, and bta-miR-1291) significantly associated with FA traits (*p*-value < 0.02, Bonferroni corrected). Among these, the rs110817643C>T, located in the seed sequence of the bta-miR-1291, was associated with different ω6 FAs, polyunsaturated FA, and polyunsaturated:saturated FA ratios. Concerning the other two miR-SNPs, the rs43400521T>C (located in the bta-miR-2419-3p) was associated with C12:0 and C18:1 cis-11 FA, whereas the rs516857374A>G (located in the bta-miR-193a-2) was associated with C18:3 ω6 and ratio of ω6/ω3 traits. Additionally, to identify potential biomarkers for FA composition, we described target genes affected by these miR-SNPs at the mRNA or protein level. Our multi-omics analysis outlines the effects of genetic polymorphism on miRNA, and it highlights miR-SNPs and target candidate genes that control beef fatty acid composition.

## 1. Introduction

Fatty acid (FA) composition is an important trait that is largely related to beef’s sensorial and nutritional properties. FA biosynthesis processes are complex and dependent on several regulatory mechanisms, such as post-transcriptional gene expression regulation [1]. However, limited knowledge of the genetic mechanisms controlling FA content and the difficulties associated with determining FA composition are restricting genetic progress related to this trait. In this context, Guo et al. [2] and De Oliveira et al. [3] have provided evidence in support of a key regulatory role of miRNAs on adipogenesis and FA composition in cattle.

MiRNAs play a pivotal role in the post-transcriptional regulation of gene expression, driving either the degradation or the inhibition of mRNA translation [4]. Since the first description of miRNAs in cattle [5], several studies have been carried out showing the impact of miRNAs on different traits such as embryonic development and implantation [6,7], prolificacy [8], growth and meat quality traits [9], feed efficiency [10,11], and beef tenderness [12].

Single nucleotide polymorphisms (SNP) can potentially disrupt miRNA expression and interaction with a target sequence. SNPs are the most common type of genetic variation, being able to explain individual differences in phenotypes [13]. The impact of a miR-SNP (SNP within a miRNA sequence) on miRNA function depends on the SNP position. For instance, miR-SNPs located in the first 14 nucleotides of the mature sequence, and mainly those located in the seed region (mostly situated at positions 2–7 from the miRNA 5’-end) may lead to novel target sites that potentially affect canonical wild-type miRNA-mRNA interactions [14,15,16]. These changes may lead to an extensive rewiring of the miRNA-mediated regulatory network and, in some instances, substantially modify a phenotype [17].

A previous report with three dairy cattle breeds has shown that miRNA genes were enriched in a genome-wide association study (GWAS) encompassing milk production traits and mastitis [18]. These findings suggest that the phenotypic variation observed may be associated with differential miRNA regulation/action. Jiang et al. [19] reported a candidate functional miR-SNP located in the seed region of the bta-miR-2899. According to the authors, this miR-SNP impairs the regulatory role of the bta-miR-2899 on the *SPI1* gene, likely contributing to predisposition of Chinese Holstein cows to mastitis. Finally, other studies have shown that cattle have more miR-SNPs than other species [20,21]. Therefore, miR-SNPs are thought to be an important tool for animal selection, potentially contributing to improving livestock traits of economic relevance. Despite this interesting scenario, both the identification of miR-SNPs and their potential role in cattle are still at an early stage. Hence, this study aimed to identify miR-SNPs in the Nelore beef cattle, their potential impact on FA composition, and functional effects on predicted target genes by taking advantage of a multi-omics approach.

## 2. Materials and Methods 

### 2.1. Animals and Phenotypic Data

Experimental procedures were carried out following the guidelines provided by the Institutional Animal Care and Use Ethical Committee of the Embrapa Pecuária Sudeste—CEUA, who approved all experimental protocols (CEUA Process 01/2013). A total of 374 Nelore steers from the Brazilian Agricultural Research Corporation (EMBRAPA) experimental breeding herd, raised between 2009 and 2011, were used. These steers were sired by 26 unrelated sires, and were selected to represent the main commercialized Nelore genetic lines in Brazil. Animals were raised in pasture and finished in feedlots under identical nutritional and handling conditions until slaughter, at an average age of 25 months, as previously described [22]. Samples from *Longissimus thoracis* (LT) muscle, between the 12th and 13th ribs, were collected at two moments: (i) At slaughter, immediately snap-frozen in liquid nitrogen, and stored at −80 °C for RNA sequencing and proteomics analysis; (ii) 24 h after slaughter, vacuum packaged and stored at −20 °C for FA composition measurement.

Description of phenotypic data and FA composition measurement were previously reported [23]. Briefly, approximately 4 g of LT muscle were lyophilized and used for FA composition determinations. Lipids were extracted for FA composition, according to the Hara and Radin [24] methodology, except for the hexane to propanol ratio being increased to 3:2. The extracted lipids were hydrolyzed and methylated according to the method described by Christie [25], except that hexane and methyl acetate were used instead of hexane:diethyl ether:formic acid (90:10:1). Fatty acids were identified by comparison of retention time of methyl esters of the samples with standards of FA butter reference BCR-CRM 164, Anhydrous Milk Fat-Producer (BCR Institute for Materials and Reference Measurements) and also with the commercial standard for 37 fatty acids Supelco TM Component FAME Mix (cat 18919, Supelco, Bellefonte, PA, USA). Fatty acids were quantified by normalizing the area under the curve of methyl esters using Chromquest 4.1 software (Thermo Electron, Milan, Italy), and expressed as a weight percentage (mg/mg). These analyses were performed at the Animal Nutrition and Growth Laboratory at ESALQ (Piracicaba, São Paulo, Brazil). The descriptive statistics of all phenotypes related to FA composition are reported in Appendix A. Variance components and genomic heritability of these traits in our population were estimated and previously reported by Cesar et al. [23]. 

### 2.2. DNA Extraction and Genotypic Data 

For sires, straws of frozen semen obtained from Brazilian artificial insemination centers were used to extract DNA by a standard phenol-chloroform method. DNA concentration was measured by spectrophotometry. The quality was verified by the 260:280 ratio, followed by inspection of integrity through agarose gel electrophoresis, as described in detail by Tizioto et al. [26]. Whole-genome sequence data of 26 progenitors of the population were obtained with the Illumina HiSeq 2500 System (Illumina Inc., San Diego, CA, USA) at 8-21x of coverage. Alignment, variant calling, and quality controls were performed according to the 1000 Bulls Genomes Project (http://www.1000bullgenomes.com/) recommendations and used as reference data for SNP imputation. Briefly, reads were trimmed and filtered using the Trimmomatic v.0.38 program [27] and then mapped to the ARS-UCD1.2 Bovine reference genome using the Burrows–Wheeler Aligner (BWA) v.0.7.17 [28]. The Samtools v.1.8 [29] was used to sort the mapped reads by sequence coordinates. Base quality recalibration (BQSR) was performed with the Genome Analysis Tool Kit (GATK version v3.8). Finally, SNPs were called using the GATK ‘HaplotypeCaller’. 

For the progeny, 5 mL of blood samples were collected and used for DNA extraction, as described in Tizioto et al. [26]. All sires and steers were genotyped using the Illumina BovineHD BeadChip 770k (Illumina, San Diego, CA, USA). Genotypes were called in the Illumina GenomeStudio software and were reported in a previous study [22,26]. BovineHD genotypes were phased using Eagle [30], and imputed using the Minimac3 program [31]. A leave-one-out cross-validation scheme was performed among the 26 sequenced individuals to assess imputation accuracy. Each sequenced animal was deleted at a time from the reference set and included as a target individual with only HD data information to be subsequently imputed with the progeny individuals. Finally, imputation efficiency was computed by comparing the imputed alleles with the alleles observed on DNA-seq data for each sire. The allelic imputation error rate was estimated as the ratio between the number of incorrect imputed alleles and the total of alleles imputed. The accuracy of imputation was considered as the correlation between the actual and imputed genotype. SNPs whose imputation accuracy in the validation was less than 0.98 and/or the allelic imputation error rate was greater than 2% were filtered out from the imputed file. Furthermore, non-informative, sexual, and SNPs with a minor allele frequency lower than 0.01 were removed from the dataset using the PLINK software [32]. After these filtering steps, a total of 4,813,664 SNPs was available for the genetic analyses.

### 2.3. Transcriptomic Data

Total RNA from muscle samples of 192 and 180 animals was used to perform the mRNA and miRNA sequencing, respectively. To extract total RNA, approximately 100 mg of frozen tissue was grounded, and the RNA was isolated using Trizol^®^ in a standard protocol (Life Technologies, Carlsbad, CA, USA). The RNA concentration and quality were evaluated in the Bioanalyzer 2100^®^ (Agilent, Santa Clara, CA, USA). 

#### 2.3.1. mRNA Expression Data

The processing and analysis of mRNA expression data were previously described [33,34]. In brief, Illumina TruSeq^®^ RNA Sample Preparation Kit v2 Guide (San Diego, CA, USA) protocol was used to generate cDNA libraries for each sample. Paired-end (PE) sequencing was performed on Illumina Hiseq 2500^®^ (San Diego, CA, USA) platform following the standard protocols. Samples were multiplexed and run in eight lanes belonging to eight sequencing flow cells to obtain 2 × 100 bp reads. Library preparation and sequencing were conducted by the ESALQ Genomics Center (Piracicaba, São Paulo, Brazil). These mRNA samples were archived on the European Nucleotide Archive (ENA) under accessions: PRJEB13188, PRJEB10898, and PRJEB19421.

Seqyclean package v.1.4.13 [35] was used to filter PE reads, which removed all reads with a mean quality under 24 and length under 65 bp, as well as the adapter sequences. Quality control (QC) of raw RNA-Seq reads was carried out with FastQC v.0.11.2 (https://www.bioinformatics.babraham.ac.uk/projects/fastqc/). The remaining sequence reads were aligned to the ARS-UCD1.2 Bovine reference genome, identified and quantified in raw counts with the STAR v.2.5.4 software [36] by using the default parameters. The alignment was performed with the—twopassMode Basic option, and the gene counts were generated using the—quantMode GeneCounts option. Gene annotation file was obtained from Ensembl database (https://www.ensembl.org/Bos_taurus/Info/Index). Considering the mRNA count reads, those that were not expressed (reads = 0) or present in less than 80% of the animals were filtered out using the *cpm* function of the edgeR package [37]. Read counts of 14,219 genes were maintained for further analysis. Potential biases due to technical variation in gene expression among samples were evaluated by applying a Principal Component Analysis (PCA) using NOISeq v.2.22.1 [38].

#### 2.3.2. miRNA Expression Data

The miRNA sequencing, QC, alignment, and quantification were previously performed as described elsewhere [39,40]. In summary, the single-end sequencing of 42 bp was carried out in the MiSeq sequencer (Illumina^®^), in 17 different lanes using MiSeq Reagent Kit v3 (150 cycles) at the Laboratory Multiuser ESALQ (Piracicaba, São Paulo, Brazil), according to the protocol described by Illumina. The miRNA samples were archived on the ENA under accession: PRJEB42280. 

The FastQC (http://www.bioinformatics.babraham.ac.uk/projects/fastqc) and FASTX (http://hannon-lab.cshl.edu/fastx-toolkit) were used to check was used to check the quality of reads. Reads with a Phred quality score lower than 28 and shorter than 18 nt were discarded. The remaining reads were subjected to the miRDeep2 software [41] as queries for sequence alignment against the ARS-UCD1.2 Bovine reference genome, with default parameters. The bovine and human mature miRNA sequences were retrieved from miRBase v. 22 [42]. The raw counts generated by miRDeep2 were processed to filter out low or not expressed miRNAs using the *cpm* function from edgeR package [37]. MiRNAs with a *cpm* value lower than 0.5 and/or present in less than 50% of the samples were removed. Read counts of 450 miRNAs were maintained for further analysis.

### 2.4. Protein Data

The processing and analysis of LT muscle’s protein data from 105 of the animals used in this study were previously performed by Poleti [43], using an integrated transcriptome-assisted label-free quantitative proteomic approach by High Definition Mass Spectrometry. In summary, peptide samples were separated using the nanoACQUITY UPLC 2D Technology system [44] and identified by Synapt G2-S High Definition mass spectrometer (Waters, Manchester, UK). For protein identification and quantification, the raw data were searched against a Nelore transcriptome database built from the RNA-sequencing data of LT muscle. Label-free protein quantification values were generated based on the Hi3 method [45]. Only proteins identified with at least two peptides present in at least 80% of the animals were considered. Raw data of 938 proteins were used for genetic analysis.

### 2.5. Retrieval of SNPs in miRNA-Related Regions 

Genomic location annotation of the imputed genotypes was performed by using SNPeff software [46], considering the ARS-UCD1.2 Bovine reference genome coordinates. After that, a total of 25 miR-SNPs located in the mature and seed regions of 22 known miRNAs were extracted by using the SNPsift software [46]. A second filter was performed to obtain only miR-SNPs in sequences of miRNAs expressed in LT muscle in our population. A total of five miR-SNPs in five miRNA sequences were considered for association analysis (Appendix A).

### 2.6. Association Analysis

The Genome-Wide Efficient Mixed-Model Association (GEMMA) software [47] was used to implement the association analyses between genotyped miR-SNPs and each of the following “phenotypic” observations: FA composition profiles, mRNA abundance, and protein abundance. GEMMA uses a mixed model approach to account for population stratification and relatedness, by calculating a genomic kinship matrix with SNP genotypes as random effects, and provides an exact test of significance. For the genomic kinship matrix generation, we used SNP markers from the Illumina BovineHD BeadChip 770k (Illumina, San Diego, CA, USA), which were not in linkage disequilibrium with the selected miR-SNPs. The following univariate mixed model was used:**y** = **Wα** + **x**β + **u** + **ε**, **u** ∼ MVN*_n_*(0, λτ^−1^**K**), **ε** ∼ MVN*_n_*(0, τ^−1^**I***_n_*),
where **y** is an *n*-vector of “phenotypic” observations (FA composition, mRNA sequencing or protein abundance data) for *n* individuals; **W** = (**w**_1_, · · ·, **w***_c_*) is an *n* × *c* matrix of covariates (fixed effects) including a column of 1s; **α** is a *c*-vector of the corresponding coefficients including the intercept; **x** is the vector of the genotypes corresponding to the set of miRNA-related SNPs; **β** is the effect size of the marker (allele substitution effect); **u** is an *n*-vector of random effects; **ε** is an *n*-vector of errors; τ^−1^ is the variance of the residual errors; λ is the ratio between the two variance components; **K** is a known *n* × *n* relatedness matrix and **I***_n_* is an *n* × *n* identity matrix. MVN*_n_* denotes the *n*-dimensional multivariate normal distribution. 

Statistical models included different batch effects for the different “phenotypic” observations (FA composition, mRNA sequencing, and protein abundance data) that were included as fixed effects in the models. Association analysis between miR-SNPs and FA composition included contemporary group classes as fixed effects (origin, birth year, and slaughter date) with 19 levels. In the association analysis considering mRNA expression data, the batch effect correction for the combination of sequencing flow cell and sequencing lane (with 22 levels) was included. Regarding the proteome data, different runs and equipment (with 5 levels) were included as fixed effects. Association analyses were assessed on the basis of the estimated allele substitution effects (β), where the alternative hypothesis H1: β ≠ 0 was contrasted against the null hypothesis H0: β = 0 with a likelihood ratio test. Bonferroni correction was applied to adjust for multiple testing, and a significance threshold was set at maximum 10% error probability.

### 2.7. Pre-miRNA Secondary Stem–Loop Structures

SNPs in miRNA sequences may also play important roles in miRNA biogenesis, impairing or enhancing miRNA processing [48]. To determine if the miR-SNP affects the pre-miRNA structure, we retrieved the pre-miRNA sequences from miRBase database v.22 [49] and analyzed the secondary stem–loop structures using the default parameters of the RNAfold web server (http://rna.tbi.univie.ac.at/).

### 2.8. miRNA-mRNA Interaction Analysis

To this end, the current annotation of the 3′UTR of bovine mRNA transcripts from Ensembl repository (http://www.ensembl.org/info/data/ftp/index.html) was downloaded and miRNA sequences were retrieved from miRBase database v.22 [49].

The analysis of loss or gain binding sites to miRNA targets was performed by the RNAhybrid [50] software to predict the target sites for the major allele-type miRNAs and miR-SNP minor allele. The RNAhybrid [50] was adopted to calculate the minimum free energy (MFE) of the miRNA–mRNA interaction setting the energy value ≤ −15 kcal/mol and *p*-value < 0.05. Approximate *p*-values were calculated considering the *3utr_human* option. For each miR-SNP, the reference and mutant allele of the mature sequence were contrasted to the 3′UTR of bovine mRNA transcripts. 

RNAhybrid predicts potential miRNA targets using an estimated MFE [50]. Mathematically, ΔMFE in kcal/mol of miRNA-mRNA interaction introducing both alleles of the miR-SNPs were calculated considering ΔMFE = MFE_major_ − MFE_minor_, where MFE_major_ is the MFE of the major allele of the miR-SNP, and MFE_minor_ is the MFE of the minor allele (effect allele associated) of the miR-SNP. ΔMFE represents the degree of miRNA regulation change from major allele-type to the minor-type. The positive value of ΔMFE demonstrates increased miRNA-mRNA stability, whereas the negative value of ΔMFE indicates the reduction of the miRNA-mRNA stability for the miR-SNP associated allele. Using the RNAhybrid predictions, gain or loss of miR-SNPs were classified to one of the four classes: (i) “Complete gain-of-function”, when the miRNA acquires a new target site with the miR-SNP minor allele; (ii) “complete loss-of-function”, when miRNA loses a predicted target site with miR-SNP minor allele; (iii) “partial gain-of-function”, when miRNA acquires more stable target site with the miR-SNP minor allele; and (iv) “partial loss-of-function”, when miRNA target site turns into unstable target site with the miR-SNP minor allele. Only genes the were significantly associated at mRNA or protein levels and whose expression change (effect size) was consistent with the hypothesis of miRNA interaction based on the estimated ΔMFE values, i.e., increased expression at mRNA or protein levels in the presence of the minor allele and negative ΔMFE, or decreased expression at mRNA or protein levels in the presence of the minor allele and positive ΔMFE were considered as target genes.

### 2.9. Pathway Analysis

The STRING network database [51] was used to generate protein–protein interaction networks for genes associated with the rs516857374A>G, to perform the functional annotation as well as to retrieve pathways and Gene Ontology (GO) functions. For that, we provided a list of gene names associated with this miR-SNP. Pathways and GO terms presenting an FDR < 0.05 were considered significantly over-represented. We did not perform pathway and GO enrichment analysis, as a low number of significantly associated genes was found for the bta-miR-2419-3p and bta-miR-1291 miRNAs.

An overview of the methodological approach and miR-SNP prioritization is summarized in Figure 1.

## 3. Results

Following SNP imputation, we identified a total of 25 miR-SNPs in the Nelore population. These SNPs were filtered considering only miRNAs expressed in LT muscle, which resulted in selecting five miR-SNPs (Appendix A). Among these, three miR-SNPs showed significant associations with FA composition traits (Table 1). 

SNPs in miRNA genes-including those that will be located in the mature region after pri- and pre-miRNA processing, could affect the stability of the secondary structure of the pre-miRNA, affecting the miRNA production. Therefore, we examined the pre-miRNA secondary structures of the three miRNAs associated with FA composition. This analysis demonstrated that the miR-SNPs were predicted to change the secondary structures of the bta-miR-2419-3p, bta-miR-193a-2, and bta-miR-1291, with the MFE of the pre-miRNA sequence ranging from -50.37 to −44.83 kcal/mol, −19.77 to −20.92 kcal/mol, and −28.76 to −27.65 kcal/mol, respectively (Figure 2).

The T allele of rs43400521T>C, located in the mature region (13th nt) of the bta-miR-2419-3p, was associated with a lower amount of C18:1 *cis*-11. Additionally, the same allele is associated with a higher content of C12:0 saturated FA (Table 1 and Figure 3). Associations between miR-SNPs in mature regions and gene expression at mRNA and/or protein levels were assessed to predict the potential effect of miR-SNPs on either disturbing or creating miRNA-target interaction sites. As a result, we found three putative target genes that had their mRNA levels affected by the T allele of rs43400521T>C miR-SNP (Figure 3 and Appendix A). The T allele of rs43400521T>C can be classified as a partial gain SNP. This allele was associated with decreased expression of both the *PNMT*-phenylethanolamine N-methyltransferase (ΔMFE= 1.8 kcal/mol; β = −8.5, *p*-value = 1.17 × 10^−2^) and the *RTN4R*-reticulon 4 receptor (ΔMFE= 0.4 kcal/mol; β = −4.0, *p*-value = 3.20E−03) genes. In addition, this allele putatively created a new miRNA-mRNA interaction site between bta-miR-2419-3p and *MFSD3*-major facilitator superfamily domain containing 3 gene, decreasing significantly its expression (β = −8.0, *p*-value = 1.00 × 10^−2^) (Appendix A).

The rs516857374A>G miR-SNP, located in the 10th nt of the bta-miR-193a-2 mature sequence, had its G allele associated with an increase of γ-linolenic acid and ω6:ω3 ratio (Table 1 and Figure 4). This allele was found to modify the binding sites of 30 putative target genes at the mRNA expression level (Appendix A). The G allele of the rs516857374A>G was predicted to create a complete gain-of-function binding site, decreasing the expression of 19 genes, e.g., *TGFBI, IGFBP6, THBS3,* and *ITGB5* (Appendix A). Partial gain-of-function was also found for other genes, with ΔMFE changes ranging from 0.4 to 2.0 kcal/mol (Appendix A). GO, analysis using the STRING network database showed that genes associated with the rs516857374A>G miR-SNP are involved in the ECM-receptor interaction and focal adhesion processes (Figure 5 and Appendix A).

The miR-SNP rs110817643C>T, located in the seed region (6th nt) of the bta-miR-1291, was associated with different FA composition traits. The T allele of the rs110817643C>T miR-SNP was associated with decreased levels of C20:3 ω6, C20:4 ω6, linoleic acid (C18:2 *cis*-9 *cis*-12 ω6), the sum of PUFA, and PUFA: SFA ratio (Table 1 and Figure 6). Protein abundance data confirmed the potential of the rs110817643C>T to modify miRNA-target interactions with two mRNA. The T allele was predicted to disrupt the interaction between bta-miR-1291 with its putative target genes, increasing the protein levels of *TCEA2* (transcription elongation factor A) and *BCAR1* (breast cancer anti-estrogen resistance 1) (Figure 7, Appendix A).

## 4. Discussion

In recent years, miR-SNPs and SNPs in target sites have been widely studied for their association with different diseases and phenotypes in humans [52]. In livestock species, SNPs in miRNA may alter miRNA processing, leading to functional alterations associated with production traits [15,19,53]. Thus, identifying functional miR-SNPs is of interest for complex trait studies. Herein, we identified miR-SNPs associated with muscle FA composition profile in Nelore cattle and performed different predictions based on potential effects on miRNA target binding affinity through a multi-omics approach. Considering the potential impact on miRNA-mediated regulation of associated traits, the predicted functional miR-SNPs should be further investigated, as they may contribute to the comprehension of these regulatory mechanisms.

### 4.1. A miR-SNP Located in the bta-miR-2419-3p Sequence Was Associated with C12:0 and 18:1-cis 11 FAs Profiles

In this study, we found that the T allele of the rs43400521T>C miR-SNP, located in the mature sequence of the bta-miR-2419-3p was associated with decreased 18:1-*cis* 11 and increased C12:0 levels of saturated FA. In silico analysis showed that the T allele of rs43400521T>C decreased the stability of this pre-miRNA with a +5.5 kcal/mol MFE change. Pre-miRNA structure can affect its turnover and function [41]. Sun et al. [48] found miR-SNPs located in the mature region, which destabilized the secondary structure, blocking the processing of pre-miRNA to mature miRNA of both strands, as well as reducing miRNA-mediated translational suppression. In this sense, rs43400521T>C miR-SNP is a candidate to destabilize the secondary structure, altering the processing of bta-miR-2419-3p and bta-miR-2419-5p. Previously, De Oliveira et al. [3] have reported an association between increased levels of the bta-miR-2419-5p and a higher content of conjugated linoleic acid (CLA-c9t11) in this Nelore population. The 18:1-*cis* 11, also called *cis*-vaccenic acid, is the FA precursor of CLA-c9t11 [54]. However, no impact of the rs43400521T>C miR-SNP in the CLA-c9t11 content could be confirmed in the association analysis. 

Association analysis suggested that bta-miR-2419-3p-SNP can impact the expression of *PNMT*, *RTN4R*, and *MFSD3* genes at mRNA levels. The T allele of the rs43400521T>C increases the MFE of miRNA-mRNA interaction, creating a new binding site in the 3′UTR sequence of the *PNMT* and *RTN4R* genes. *PNMT* is a protein-coding gene that catalyzes the final step in epinephrine biosynthesis-one of the major hormones involved in glucose counter-regulation and gluconeogenesis [55,56]. Sharara-Chami et al. [56] showed that *PNMT* may play an important role in FAs oxidation control. In addition, Gomes et al. [57] demonstrated that unsaturated FAs affected catecholamines handling through decreased o *PNMT* expression level. This finding suggests that the amines might indeed constitute mediating factors in the relationship between unsaturated FAs and metabolic syndrome.

The *RTN4R*, known as Nogo receptor 1, may play a role in regulating axonal regeneration and plasticity in the adult central nervous system [58]. However, to the best of our knowledge, no direct effect of *RTN4R* on FA metabolism has been described so far. However, *RTN4R* expression is necessary to the endoplasmic reticulum (ER) formation and stabilization [59], a known place for FA elongation and biosynthesis [60]. Finally, we showed that the T allele of the rs43400521T>C creates a complete miRNA-mRNA interaction between the miR-2419-3p and the *MFSD3* gene. According to the Kyoto Encyclopedia of Genes and Genomes [61], the *MFSD3* encodes for a putative acetyl-CoA transporter, which has a relatively high sequence identity with the *SLC33A1*, a known acetyl-CoA transporter [62]. SLC33A1 is a key regulator in intracellular acetyl-CoA homeostasis in the ER and can act as a metabolic regulator, including reprogramming of lipid metabolism and mitochondria bioenergetics [63]. In addition, Palombo et al. [64] have associated polymorphisms in the *MFSD3* gene of cows with saturated and monosaturated FA composition in milk, providing a possible functional implication of the rs43400521T>C.

Saturated medium-chain FAs, such as the lauric acid (C12:0), are more effectively absorbed and metabolized than saturated long-chain FA. Additionally, C12:0 is the most potent antimicrobial saturated FA [65,66]. Food enriched with medium-chain FA can increase the ketone content, positively impacting the ratio between total and high-density lipoprotein cholesterol [67,68]. Hwang and Joo [69] have shown that high-fat and high-marbled muscles, such as LT, have a higher proportion of C12:0 that is positively correlated with sensorial traits. In the present study, we showed that the miR-2419-3p-SNP was associated with C12:0 profile, which is estimated to have a low heritability in our Nelore population [21]. Thus, this miR-SNP is a candidate biomarker to be included in dense SNP chips along with other FA-associated SNPs.

### 4.2. A bta-miR-193a-2-SNP May Influence γ-Linolenic Acid and ω6/ω3 Ratio Profiles 

The rs516857374A>G miR-SNP, harbored in the 10th nt of the bta-mir-193a-2 mature sequence, had its G allele associated with an increase γ-linolenic acid (G-LNA) content and ω6/ω3 ratio. Previous studies have implicated the miR-193a on regulatory networks of human adipose tissue and obesity [70]. Zhang et al. [71] explored miRNAs and pathways regulating intramuscular fat (IMF) deposition. They found the bta-miR-193a-3p differentially expressed in IMF tissues and present in molecular networks functionally associated with adipocyte differentiation and adipose tissues metabolism. G-LNA is found in small amounts in various common foods, notably organ meats and milk [72]. Numerous in vitro and in vivo studies have demonstrated that G-LNA-supplemented diets attenuate different inflammatory responses (reviewed by Sergeant et al. [73]). Furthermore, ω3 PUFAs are known to have anti-inflammatory properties, and the ratio ω6/ω3 is an important FA parameter for chronic diseases and inflammatory processes [74]. 

The *CCDC80* (Coiled-Coil Domain Containing 80) was shown as a central core of the network predicted by STRING, and showed a decrease of its expression at mRNA level in response to the presence of the G allele of the rs516857374A>G. Recently, Li et al. [75] showed that *CCDC80*-knockout could down-regulate *PPAR* signaling and fatty acid degradation. Furthermore, we identified that the G allele of the rs516857374A>G creates binding sites to a total of 30 genes. Overall, our result suggests that this allele can control genes involved in ECM-receptor interaction and focal adhesion. Both processes are formed by a complex network of different proteins and proteoglycans that control cell adhesion and signaling associated with obesity and metabolic diseases [76,77]. Cesar et al. [34] identified ECM-receptor interaction as a functional enrichment term associated with IMF content traits in the same population studied here. Likewise, Diniz et al. [33] reported the *ITGB5* as a hub gene present in a network-module associated with IMF. Altogether, our results provided new insights into the relationship between cell adhesion genes and FA composition in Nelore cattle.

### 4.3. Seed SNP in the bta-miR-1291 Controlling the Composition of Many ω6-PUFA FAs 

The T allele of the rs110817643C>T, in the bta-miR-1291 seed region, was significantly associated with decreased content of C18:2 *cis*-9 *cis*-12 ω6, 20:3 ω6, and C20:4 ω6, resulting in a reduction of the sum of PUFA and the ratio of PUFA to SFA. De Oliveira et al. [78] have investigated regulatory candidate genes and co-expression networks related to IMF content. These authors identified the bta-miR-1291 as a hub miRNA in the low IMF group, in the same Nelore population assessed here, demonstrating the impact of the bta-miR-1291 in fat deposition. Braud [79] investigated polymorphisms in miRNAs in different cattle breeds. They reported the same SNP identified by us in the bta-miR-1291 as shared by all analyzed breeds and influencing genes present in QTLs related to many FA traits, such as ω6/ω3 FA ratio, as well as milk linoleic acid percentage. These findings suggest that this candidate miR-SNP can be relevant for further studies using other experimental populations and breeds aiming to improve the FAs composition of bovine muscle. 

In the presence of the C allele of the rs110817643C>T, there is a putative interaction between bta-miR-1291 and *TCEA2*, and between bta-miR-1291 and *BCAR1*. However, as the only significant predicted bindings were found for the C allele, the T allele is a candidate for disrupting the binding site of the bta-miR-1291 with these genes. Both genes, *TCEA2* and *BCAR1*, showed an increase of protein abundance associated with the T allele of the rs110817643C>T miR-SNP, corroborating with the in silico binding site prediction. The *TCEA2*, also known as *TFIIS*, is necessary for efficient RNA polymerase II transcription elongation [80]. Downregulation of the *TCEA2* expression was previously related to 20:4 ω6-rich diet compared to a ω3-rich diet [81] in murine liver. Furthermore, *TCEA2* was identified as a candidate gene involved in hypertension (Cao et al., under revision, 2020), a disease commonly associated with ω6 FA [82]. *BCAR1* was previously identified as a candidate regulatory gene of IMF deposition and FA content in cattle and sheep [33,34,83]. In our Nelore population, Diniz et al. [33] identified the presence of the *BCAR1* gene in a module network negatively associated with IMF. While Cesar et al. [34] found a negative correlation between the co-expression modules that contain the *BCAR1* gene and IMF, linoleic acid, and ω6 sum.

ω6-PUFA and molecules derived from them, including linoleic acid- and arachidonic acid-derived lipid mediators, are known to have pro- and anti-inflammatory properties [84]. On the other hand, SFA can promote inflammation by increasing the secretion of pro-inflammatory cytokines [85]. Meat is an important source of fat, and the ideal balance of PUFA:SFA is 1:1 for generating the best LDL/HDL ratios in a diet [86]. Additionally, the beneficial effects of PUFA depend on the ω6/ω3 ratio; when the ideal proportion is between 2:1 and 1:1 [87]. In our animal population, Cesar et al. [23] described a balance of 0.06 and 1.44 for PUFA:SFA and ω6/ω3 ratios, respectively. If the association between this miR-SNP in the bta-miR-1291 and FA composition is validated in other populations of Nelore, this information could be used to improve accuracy of SNP panels applied to FA traits’ genomic selection.

Our study used different layers of gene regulation (i.e., proteome and transcriptome) to analyze miR-SNPs’ impact on its miRNA-target genes. These layers are dynamic in nature, and their cross talk is overwhelmingly complex [88], and measurements taken from mRNA and protein levels are complementary [89]. Unfortunately, it was impossible to identify the abundance at protein levels for all candidate genes associated with rs43400521T>C and rs516857374A>G at the mRNA level. Our proteomics data (938 proteins) compared to mRNA-sequencing data (14,219 genes) were limited, because we restricted the dataset to proteins that are safely quantified, i.e., those of greater abundance and present in 80% of the samples.

Binding of a miRNA to a mRNA either triggers mRNA cleavage and decay or inhibits translation, predominantly without degrading the mRNA [90], thus a multi-omic approach may help indicating the mechanism involved in a specific miRNA-gene interaction. This was the case of *TCEA2* and *BCAR1* genes, which showed significant reduction at protein level associated with the C allele of rs110817643C>T miR-SNP, but absence of significant association at the mRNA level. Thus, our results suggest the translation inhibition as the probable mechanism of bta-miR-1291 regulation of *TCEA2* and *BCAR1* expression. It demonstrated that different levels of gene expression data are essential to explain genotype–phenotype relationships and provide new insights for the understanding of biological processes.

## 5. Conclusions

In this study, we systematically examined the association of miR-SNPs with FA profiles in Nelore cattle muscle by using genetic and genomic information. Based on our approach, three miR-SNPs, located in the bta-miR-2419-3p, bta-miR-193a-2, and bta-miR-1291 were shown to be associated with C12:0 and C18:1 *cis*-11 FA, C18:3 ω6 and ω6/ω3 ratio and, different ω6 FAs, PUFA and PUFA:SFA ratio, respectively. Furthermore, we showed the putative impact of these miR-SNPs on the miRNA-mRNA interactions, and evidenced their consequent effect on gene expression at the mRNA and protein levels. Future experimental studies are needed, however, to elucidate the mechanisms underlying the link between these miRNAs and their putative targets in determining phenotypes of economic interest in Nelore cattle.

## Figures and Tables

**Figure 1 genes-12-00067-f001:**
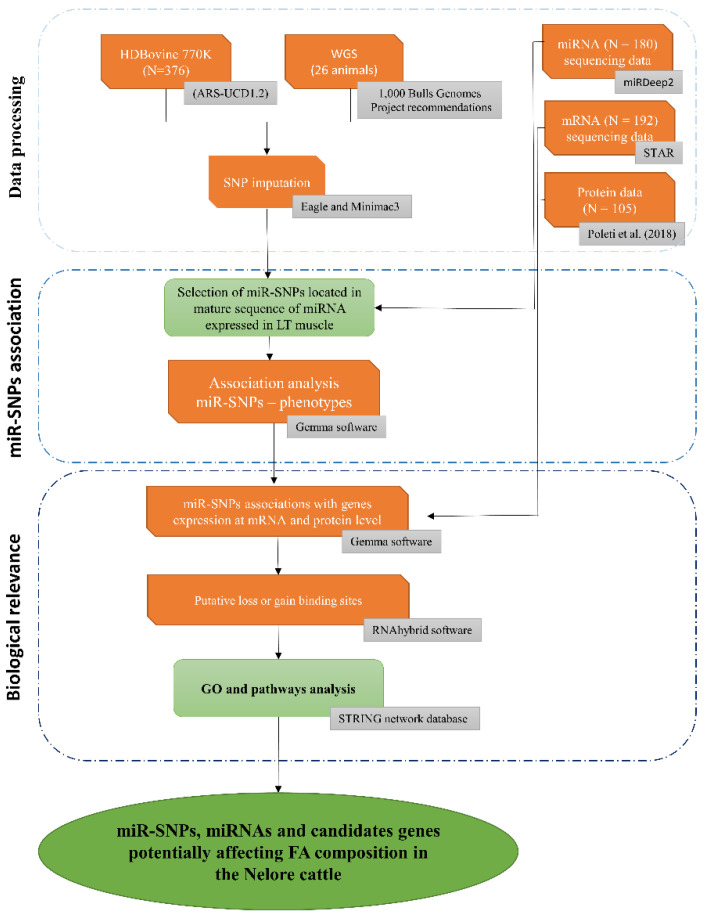
Overview of the multi-omics analysis approach for identification of polymorphisms in miRNA mature sequence associated to fatty acid composition in the Nelore cattle.

**Figure 2 genes-12-00067-f002:**
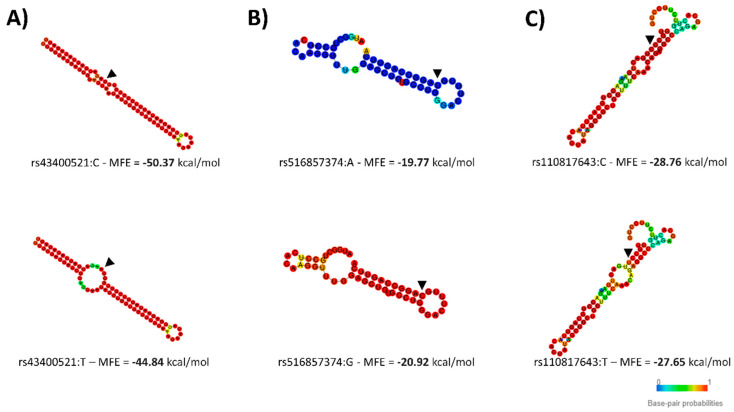
In silico analysis of potential impact of miR-SNPs. Secondary structure predicting the impact of the miR-SNPs in pre-miRNAs sequences: (**A**) Secondary structure of bta-mir-2419 with the rs43400521:T>C miR-SNP, (**B**) Secondary structure of bta-mir-193a-2 with the rs516857374:A>G miR-SNP and (**C**) Secondary structure of bta-mir-1291 with the rs110817643:C>T miR-SNP. The secondary structures of the pre-miRNAs were predicted by inputting two transcript sequences, corresponding to both alleles of the miR-SNPs, i.e., either the major (top) or minor (bottom) allele submitted to RNAfold. Figures and minimum free energy (MFE) values were generated by RNAfold.

**Figure 3 genes-12-00067-f003:**
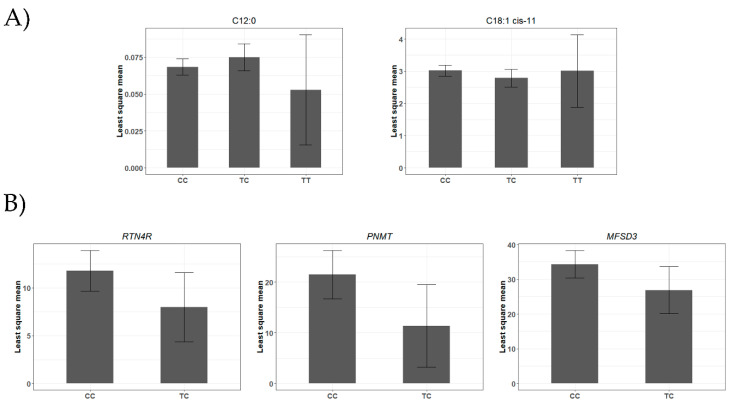
Association analysis of rs43400521T>C miR-SNP genotype in the bta-miR-2419-3p. (**A**) Least square means for the fatty acids associated traits identified by Gemma software (CC, *n* = 326; TC, *n* = 46; and TT, *n* = 2). (**B**) Least squares means for the genes associated identified by Gemma software (CC, *n* = 164 and TC, *n* = 28).

**Figure 4 genes-12-00067-f004:**
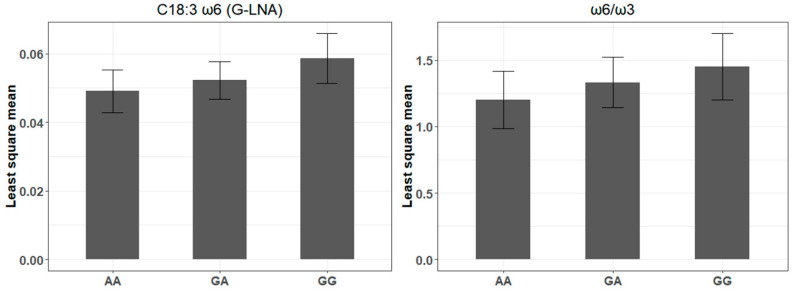
Association analysis of rs516857374A>G genotype in the bta-miR-193a-2. Least square means for the fatty acids associated traits identified by Gemma software (AA, *n* = 130; GA, *n* = 182; and GG, *n* = 62).

**Figure 5 genes-12-00067-f005:**
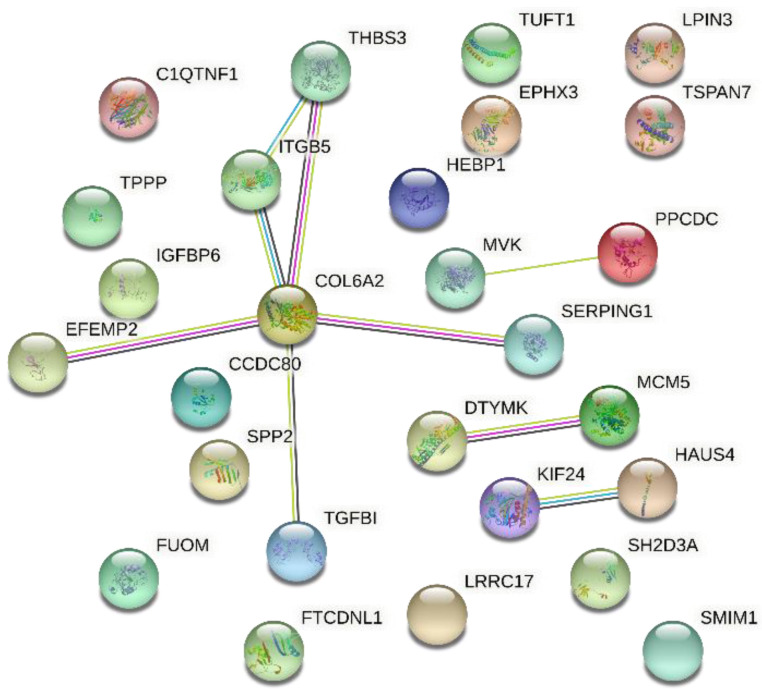
String network showing the genes whose expression decreases in response to G allele of the rs516857374A>G located in the bta-miR-193a-2 mature sequence. The network nodes are proteins and the edges represent the predicted functional associations. Each colored line represents different evidence for each interaction (red: fusion; green: neighborhood; blue: co-occurrence; purple: experimental; yellow: text mining; light blue: database; black: co-expression).

**Figure 6 genes-12-00067-f006:**
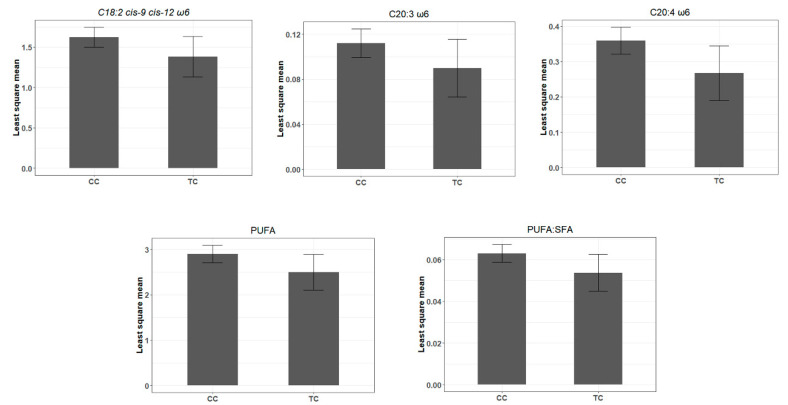
Association analysis of rs110817643C>T genotype in the bta-miR-1291. Least square means for the fatty acids associated traits identified by Gemma software (CC, *n* = 345; and TC, *n* = 29).

**Figure 7 genes-12-00067-f007:**
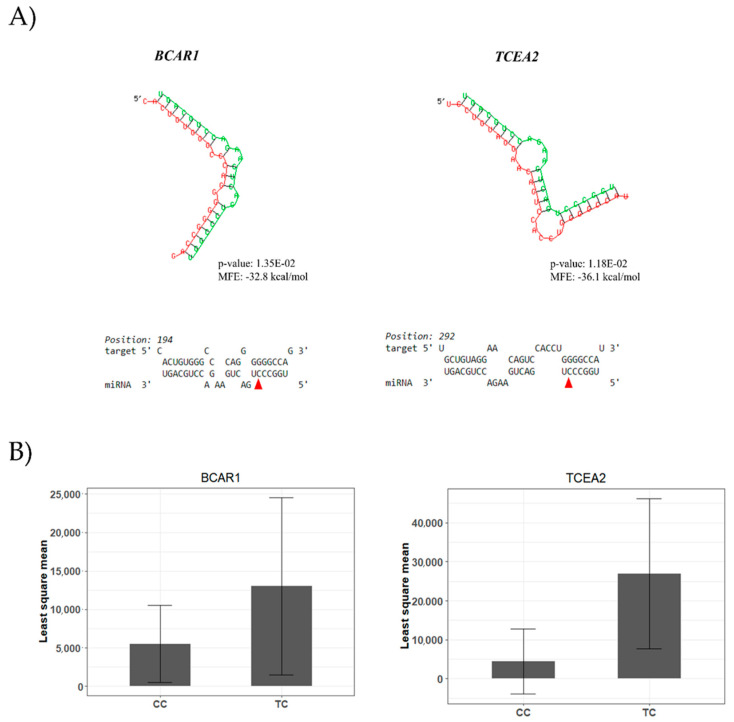
(**A**) RNAhybrid analysis of C allele of the rs110817643C>T located in the seed sequence of the bta-miR-1291 (red sequence) and corresponding target sequences on *TCEA2* and *BCAR1* 3’UTR (green sequence). The red triangle marks the SNP positions. These interactions were predicted to be lost, in both target genes, for the miRNA sequence corresponding to the T allele of the rs110817643C>T (*TCEA2*-rs110817643:T-MFE = −30.8, *p*-value = 0.08; *BCAR1*-rs110817643:T-MFE = −28.5, *p*-value = 0.07). (**B**) Least squares means by genotype for rs110817643C>T miR-SNP located in the bta-miR-1291. Least square means for the proteins associated identified by Gemma software (CC, *n* = 90; and TC, *n* = 15).

**Table 1 genes-12-00067-t001:** miR-SNPs located in mature sequences of miRNAs associated with fatty-acid composition in Nelore cattle muscle.

miR-SNP	Bp *	miRNA	MAF	Phenotype	β	SE	*p*-Value
rs43400521**T**>C	13	bta-miR-2419-3p	0.07	C12:0	8.16 × 10^−3^	3.41 × 10^−3^	1.68 × 10^−2^
				C18:1 *cis*-11	−4.13 × 10^−1^	1.72 × 10^−1^	1.72 × 10^−2^
rs516857374A>**G**	10	bta-miR-193a-2	0.41	C18:3 ω6 (G-LNA)	7.28 × 10^−3^	2.03 × 10^−3^	5.27 × 10^−4^
				ω6/ω3	1.94 × 10^−1^	7.13 × 10^−2^	7.26 × 10^−3^
rs110817643C>**T**	6	bta-miR-1291	0.04	C18:2 *cis*-9 *cis*-12 ω6	−2.81 × 10^−1^	1.03 × 10^−1^	6.24 × 10^−3^
				C20:3 ω6	−2.88 × 10^−2^	1.13 × 10^−2^	1.12 × 10^−2^
				C20:4 ω6	−9.44 × 10^−2^	3.18 × 10^−2^	3.10 × 10^−3^
				PUFA	−5.05 × 10^−1^	1.67 × 10^−1^	2.61 × 10^−3^
				PUFA:SFA	−1.25 × 10^−2^	3.97 × 10^−3^	1.79 × 10^−3^

* Position in the mature sequence. The minor allele (in bold) is the effect allele. MAF = minor allele frequency; β = allelic effect; SE = standard errors for β; C12:0 = Lauric acid; C18:1 *cis*-11= *cis*-11-Octadecenoic acid (*cis*-vaccenic acid); C18:1 *cis*-12 = *Cis*-12 Octadecenoic; C18:2 *cis*-9 *cis*-12 ω6= Linoleic acid; C18:3 ω6 (G-LNA) = γ-Linolenic acid; C20:3 ω6 = Dihomo-γ-linolenic acid; C20:4 ω6 = Arachidonic acid; ω6/ω3 = Ratio of omega-6 to omega-3; PUFA = Sum of polyunsaturated FA; PUFA:SFA = Ratio of PUFA to saturated FA.

## Data Availability

The datasets generated for this study can be found in the European Nucleotide Archive (ENA)/PRJEB13188, PRJEB10898, PRJEB19421, and PRJEB42280.

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
