# Peer review of "Multi-Omics Approach Reveals miR-SNPs Affecting Muscle Fatty Acids Profile in Nelore Cattle"

_genes, 2021, doi:10.3390/genes12010067_

Round 1

Reviewer 1 Report

The authors in this manuscript endeavoured to investigate molecular genetic architecture of variability in fatty acid content in the Longissimus thoracis skeletal muscle. They explored association of SNPs located in miRNAs with Fatty acids and investigated the potential impact of these miRNA-SNPs on target-miRNA interactions. It is such an interesting study that has potential to contribute to our understanding of the genetic background of fatty acid content in beef. The manuscript is generally well structured however, there is a lo of improvement required before it can be accepted for publication. The paper needs grammar review, the language does not flow, probably a professional scientific English editor would potentially improve the quality of the manuscript. The main problem with the manuscript is the materials and methods part. The authors over summarized what they did or what really happened through the experiment (Currently, there is a lot of missing important information in the materials and methods), this will make it hard for any scientist who would wish to reproduce the experiment, my suggestion is that the authors extensively devote time to revise their materials and methods, describing their methodology well to someone reading their paper.

Line60: Add “that is” before “largely”.

Line64-65: Please rephrase this “In keeping with this,”

Line71: DOI: 10.1038/s41598-020-73885-5 also investigated involvement of miRNAs in feed efficiency in beef cattle.

Line75: Remove “however”

Line78-80: Any reference for this information?

Line84-85: “In turn” does not seem a suitable linking phrase as Jiang et al (2019) is not a direct consequential study from Fang et al(2018). Please rephrase with a better phrase.

Line91: Replace “yet” with “still” or rephrase the whole sentence as it doesn’t sound right.

A general observation, please avoid assuming all your readers know what exactly you did and how you did it, try to provide information that would help your readers follow your methodology without having to often look somewhere else to figure out what you did or how you did it.

Was your experiment reviewed and approved by an established ethical committee for animal experimentation?

Line97: Change “of 374” to “from 374”

Line97: Add “which were” before “offspring”

Line98-99: It is fine to give a citation for the detailed description of the experiment, however you should give a brief description production, slaughtering (e.g. Average slaughter age etc.) here, not just giving the citation.

Line100-101: Please describe briefly what you did measuring FA content before citing the detailed account.

Line104-105: Please state precisely, if the sample for miRNA and mRNA and proteomics was taken from the same tissue as that for FA measurement.

Line107: Please help your readers and describe briefly what was done, before citing another study for details.

Line110-111: Please describe what you did in the alignment and SNP calling, what tools did you use for this purpose.

Line112-113: Please start with “For the progeny, 5 ml of blood..” This could actually start a new paragraph.

Line117-118: How did you test for imputation accuracy? Include it in your description.

Line123: Has the sequence data been submitted to a public repository such as GEO, if so state the accession number.

Line124-125: Please give a brief description of what you did to extract the mRNA and miRNAs, and preparation of both cDNA libraries.

Describe the sequencing process of both for mRNA and miRNA, how many flow cells used and how many samples per lane on the flow cell. That information is needed, you should not just give few highlights of what really happened in the experiment. What kind of reads did you generate for mRNA sequencing, paired or single-end?

Before aligning please describe what you did to raw sequence reads in terms of quality assessment, and quality control cleaning. What tools did you use and what was removed and what was considered clean?

Line125-133: Please do not try to compress many things into one sentence (you done this many times in the manuscript). One can hardly follow what you exactly did with both mRNAseq and miRNAseq data. You over summarised what you did, for example, STAR is used as an aligner but doesn’t count feature individual alignment. Please rewrite this part step by step providing all the tools you used and the parameters you used. You could review how other people who have done similar studies describe their bioinformatic analysis of the data of both miRNA and mRNA data.

Write detailed description of how you worked with both miRNA and mRNA datasets independently and step by step, not mixed up together. How did you clean the miRNA data.

Line132-133: How do you retain 14219 mRNA and 450 miRNA sequences? This is confusing, probably you wanted to mean 14219 genes and 450 miRNAs.  As mentioned above please rewrite this part giving details of what you did, step by step, to help your readers follow you through your description.

Line134-141: Please revise this section providing detailed description of what you did step by with precision (From Extraction to quantification) not just general description. Please look at this article (https://doi.org/10.1038/s41598-020-59412-6) and you should consider presenting your materials and methods in the same format.

Line143-144: Why did you only extract SNPs from only 22 known miRNAs? What was the rationale, please provide this in the description?

L154: what do you mean by unrelated genotypes? Probably you wanted to mean SNP markers in low LD or non associated SNP marked. Please rewrite.

L157-158: What fixed effects did you have in your model?

Line163-164: what do you mean by this “Statistical models included different batch effects for different traits.” Were these added as fixed effects or what?

Line164-165: What do you mean by this “For FA composition was included contemporary group classes (the origin, birth year, and slaughter date) as fixed effects.”?

Please familiarize with your modelling and rewrite the description of your modelling, what you have written is confusing and hard to follow, which effects you included as fixed effects.

Line165: You can not use “for” and “to” at the same time.

Line165-168: What do you mean by “the same model was used” and then you describe something additional in the proceeding sentence. My view is that you are trying to over summarize your materials and methods description which is very confusing to readers, each sentence should tell a clear message and there should be flow in your description.

Line167-168: Did you test if the flow cell or sequencing lane had effect on the abundance of mRNAs? Please provide this information otherwise you might just be overfitting the model. Secondly why didn’t you do the same for miRNAs?

Line169-172: This sentence is too long. Break it down and be clearer in what you are trying to communicate.

Line170: It is “major” not “mayor”.

Line173-174: Change this sentence to something like this, Bonferroni correction was applied to cater for multiple testing, and a significance threshold set at less 10% error probability.

Line187-188: Why you refer to MFEmajor as the MFE of the major allele-type miRNA, and then you refer to MFEminor as the MFE associated to allele-type miRNA sequence. Please rewrite and keep consistent.

Line189: What do you mean by “from major allele-type to the variant-type”? what is the variant allele (Line186) and variant-type?

Line196-198: “It was only  considered target-genes those whose change of expression (effect size) was consistent with the direction of the ΔMFE values due to miR-SNP.” This does not sound grammatically fine, please revise.

Line199-200: Why did you have to analyse the miRNA secondary structure? You evaluated the stability of the secondary structure of pri-miRNA. This should probably come before target gene prediction. Please revise.

Line202-204: Please provide a detailed description of what you did and how you did it. Was the protein-protein interaction network for targets each miRNA independently or you did it for all targets for all the miRNAs.

Line207: “taking into account their expression in LT muscle” what do you mean by this? Can SNPs be expressed? Please revise this.

Line209-210: This should be in the materials and methods.

Line223: Hairpin structure of what? Revise by adding “of the pri-miRNA”.

Line224-225: You examined the secondary structure of the pri-miRNAs of the three miRNAs. Please familiarize with miRNA biology and revise your writing.

Line227: “Pre-mature” should be “pri-miRNA”, please revise.

Line229-261: Could you please add figures (bar plot) of the least square means of the significantly associated Fatty acids, for three genotypes for each of the three miR-SNPs, would be good to see the same plots for the target mRNA expression and protein abundance as well.

Line245-250: Is this all caption for Figure 2? Please separate the caption from main text for the article.

Line267: What do you mean by this “interaction of this miR-SNP with its targeting mRNA”? Does a miR-SNP have a target mRNA? Please revise your writing.

Line273-277: Is this all a caption of Figure 3, I guess not, please separate the caption from the main text.

Line273-274: What do you mean by a gene being negatively associated to a genotype of a SNP?

Line275-277:  What is the meaning of this sentence?

Line277: In Line204, you stated that you analysed for pathways and Gene ontology terms, where are the results from this analysis?

Line279-283: Is this all a caption for Figure 4? Please separate figure captions from the main text and then you can reference the Figures.

Line280-281: Should be “SNP” position not “allele position”.

Line290-292: Revise this sentence, it does not sound right.

Line292-293: What do you mean by this “Considering the wide regulation of miRNA”? Please revise.

Line297-299: You have repeatedly written that an allele of a SNP is associated to a trait in your manuscript, but this is not statistically right. SNP as a factor is the one that is associated to independent variable. So when you write that the T allele is negatively or positively associated to a given FA acid I get confused please use the right terminologies, Revise. You can say, “ The T allele is associated to decrease or lower amount of 18:1-cis, whereas the same allele is associated with increased or higher content of C12:0”

Line313-316: How is the gene PNMT related to fatty acids? Any known possible link if none you can state, some other researchers could pick on that for further studies?

Line335-337: Do you think this single marker can for potentially a complex trait like fatty acid content?

Line360: Change “..on the” to “in the”.

Line360-361: This is how you should be reporting results in the manuscript.

Line365-368: Did Braud[68] Identify the same SNP? I do not think so according to your description of the results from that study, I therefore do not agree with your conclusion (Line368-370) unless you are referring to their SNP and not yours.

Line371-375: I find it hard to follow this part, firstly, you the SNP rs110817643 is an C>T SNP, where does G, come from unless you are talking of a different strand which I think you aren’t.

Line374-375: Secondly, how can expression of genes have a positive association with the miR-SNP? I guess you are trying to directly interpret the βs, I do not think this way of interpretation is right.

Line390-392: Revise this. A SNP can be a potential candidate to be selected for FA content improvement? It doesn’t sound right.

In all your discussion I have not read discussion of the protein-protein interaction results from STRING, not sure why.

Why would a SNP be associated to the mRNA abundancy but not protein abundance of the same gene? I would expect such in the discussion of your results. Please take look at those genes that show differing association i.e. where the SNP is associated to mRNA expression level and not the protein abundance and vice versa, this might highlight on what exactly is the regulatory mechanism of the target gene by miRNA.

What do you mean by “associated with different FA profiles”? what are those different profiles? Maybe better you fatty acids, that’s enough than adding profiles, as that gives a different meaning to the sentence.

What you call Supplementary file S1-S4 are tables in one supplementary file, please revise this in your manuscript and rename the Tables as Supplementary Table S1, S2 etc in the Supplementary file 1 (if you decide to call the whole supplementary file Supplementary file 1).

Review your punctuations, you have commas in places where they are not required.

Reviewer 2 Report

The manuscript “Multi-omics approach reveals miR-SNPs affecting muscle fatty acids profile in the Nelore cattle” submitted to Genes journal is focused on a very interesting subject and aimed to identify and integrate miRNA-SNPs in the Nelore cattle individual and their potential impact on fatty acid composition as one of the economic traits with consequent effect on gene expression at the mRNA and protein levels. Using muli-omics approach and integration of different omics datasets provide stronger evidence that could support the use of specific biomarkers in breeding programs and the selection for the desired economic traits. The experiment is well designed and the paper is written in a comprehensive and concise way.

I only have a few minor remarks:

  • Line 209: The place of this sentence “An overview of the methodological approach and miR-SNP prioritization is summarized in Figure 1” is not appropriate in this section. It will be better to move this sentence to the MM section.

  • Line 262: The miR-SNP rs110817643C>T, located on the seed region of the bta-miR-1291, was not presented in the Supplementary Table S3. Is that mean this miR-SNP was not associated with the gene expression data at the mRNA level? And only associated with the protein abundance data? The authors mentioned that there is a potential of this miR-SNP to modify miRNA-target interactions with two genes (Line 266). Did the mRNA expression data confirm this modification? Did it show non-significant differences or opposite patterns? Please comment and clarify this point.
  • The other two miR-SNPs did not exhibit an association with the protein abundance data, although both were associated with the mRNA expression data. Please add a sentence to the discussion to explain this dissociation between both mRNA and protein expression datasets.

  • Did the authors deposit the raw and analysed sequencing data (miRNA and mRNA) into any data repositories (ex. Gene Expression Omnibus (GEO) database)? Please include a link or accession number that allows accessing the data.

  • Line 269: This should be Supplementary file S5, not S4.

  • Lines 371 and 373: Do you mean “C allele” not “G allele”?

Round 2

Reviewer 1 Report

I am satisfied with the changes made by the authors to the manuscript, I accept this paper for publication, though the authors should do a thorough review of the spellings and grammar of the manuscript.